# Hepatitis E Virus (HEV) Synopsis: General Aspects and Focus on Bangladesh

**DOI:** 10.3390/v15010063

**Published:** 2022-12-24

**Authors:** Asma Binte Aziz, Joakim Øverbø, Susanne Dudman, Cathinka Halle Julin, Yoon Jeong Gabby Kwon, Yasmin Jahan, Mohammad Ali, Jennifer L. Dembinski

**Affiliations:** 1Institute of Clinical Medicine, University of Oslo, 0315 Oslo, Norway; 2International Vaccine Institute (IVI), Seoul 08800, Republic of Korea; 3Norwegian Institute of Public Health (NIPH), Division of Infection Control and Environmental Health, 0213 Oslo, Norway; 4Carnegie Mellon University, Pittsburgh, PA 15213, USA; 5Graduate School of Biomedical and Health Sciences, Hiroshima University, Hiroshima 734-0046, Japan; 6Johns Hopkins Bloomberg School of Public Health, Baltimore, Maryland, MD 21205, USA

**Keywords:** hepatitis E virus (HEV), Bangladesh, vaccine, epidemiology, endemic zones

## Abstract

HEV is the most common cause of acute hepatitis globally. This review summarizes the latest knowledge on the epidemiology, clinical characteristics, testing, and treatment of HEV infection. We also focused on Bangladesh to highlight the distinct challenges and the possible remedies. In low-income settings, the virus is mainly transmitted between people by fecal contamination of drinking water causing large outbreaks, and sporadic cases. The disease is usually mild and self-limiting acute hepatitis. Still, pregnant women and their offspring in low-income countries are at particular risk for severe disease, with up to 20% maternal mortality. Despite the high burden of the disease, HEV remains a relatively neglected virus, with detection hampered by costly tests and a lack of suitable treatments. Molecular PCR diagnostics, together with ELISA antibody tests, remain the preferred methods for diagnosis of HEV; however, rapid bedside diagnostics are available and could offer a practical alternative, especially in low-income countries. One vaccine (HEV 239) is only available in China and Pakistan, as efficacy against the other genotypes remains uncertain. The effectiveness trial conducted in Bangladesh might lead the way in gathering more efficacy data and could, together with improved surveillance and raised awareness, dramatically reduce the global burden of HEV.

## 1. Introduction

Hepatitis E virus (HEV)—officially known as *Paslahepevirus balayani* (family *Hepeviridae*; subfamily *Orthohepevirinae*; genus *Paslahepevirus*) by the International Committee on Taxonomy of Viruses [1]—was named after Russian virologist Mikhail Balayani. who discovered this virus [2]. HEV is the fifth human hepatitis virus (Hepatitis A, B, C, D, and E) and the most common cause of acute hepatitis globally [3]. Considering the increased global prevalence, HEV is a significant public health concern causing an estimated 20 million infections in a year globally, with about 3.3 million symptomatic cases [4]. During 2005. the annual disease burden of HEV was modeled across the nine global burdens of disease project regions in Africa and Asia (East Asia, South Asia, Southeast Asia, North Africa, Middle East, Central Sub-Saharan, East Sub-Saharan, Southern Sub-Saharan, Central Sub-Saharan). In these regions, it was estimated that HEV is responsible for 70,000 deaths and 3000 stillbirths [5]. In 2015, HEV caused approximately 44,000 deaths, accounting for 3.3% of viral hepatitis-related deaths [4]. However, the actual burden of HEV has yet to be determined. In a country like Bangladesh, with over 170 million population, HEV is the major cause of hospitalization for acute jaundice, resulting in considerable disease and death. Despite this, HEV testing is still rare, and patients often use treatments from traditional healers in rural areas, which leads to maltreatment and dreadful complications [6]. There is a lack of reliable surveillance statistics on the burden of HEV here, like in many other endemic countries. However, an estimated nationwide HEV seroprevalence of 20% (95% CI:17–24%) was reported recently [7]. Previous cross-sectional studies indicated HEV seroprevalence to range from 22% in rural areas to 60% in the capital of Dhaka [8]. Although the case fatality rate is 5% among hospitalized HEV patients in urban and rural areas in Bangladesh, the fatality rate is much higher among pregnant women (12%) [9]. In this article, we comprehensively review the literature to give an overview of the history, taxonomy, transmission, epidemiological aspects, clinical manifestations, and diagnosis and treatment options available to combat this emerging disease, focusing on Bangladesh.

## 2. History of Hepatitis E

HEV was first recognized in November 1978 in a rural part of India known as the Gulmarg region in Kashmir after a massive water-borne jaundice outbreak. In this outbreak, two hundred villages with a population of 600,000 were affected, and among them, 52,000 developed jaundice and 1700 died. The disease was initially classified as “epidemic non-A non-B hepatitis,” a waterborne disease such as hepatitis A common in poorer countries and infrequent in developed parts of the world [10]. Later, another epidemic of hepatitis E identical to the Kashmir outbreak was reported among Russian soldiers working in Afghanistan. In 1981, Mikhail Balayani self-experimented by ingesting pooled stools of infected soldiers and developed severe acute hepatitis with jaundice. When he examined his stool samples, HEV was visualized [2]. Retrospectively, it is now confirmed that another epidemic in Delhi in 1955–1956 was also due to HEV; still, it was not recognized until after the outbreak at the Russian military camp mentioned above [11].

In 1991, the complete genetic material of HEV cDNA was first successfully cloned, and diagnostic assays were developed [12]. Subsequently, in 1997 the zoonotic potential of swine HEV (the first animal strain) was demonstrated [13]. In 2010, it was revealed that HEV also existed between the 7th and 15th centuries and in the 19th and late 20th centuries [14]. This evidence indicates that Hepatitis E is an ancient disease with historical significance [14].

### HEV Outbreaks in Bangladesh

We found six records of HEV outbreaks in four locations in Bangladesh (Figure 1). The first documented outbreak in Bangladesh was in late 2008 in the capital city (Dhaka); 4751 suspected HEV cases were identified, and 17 deaths occurred, including four pregnant women; anti-HEV IgM was identified in 77% of patients who were neighbors of the case-patients who died [15]. Then, in 2013, at Noakhali Police Training Centre, among 112 suspected cases, 76% were HEV [15], and two outbreaks were reported in northern Bangladesh, named Rajshahi city, during 2010 with HEV-1 (2162 suspected cases; from 62 probable cases, 58% were IgM anti-HEV positive) [16] and in 2017 [15]. There were two epidemics in the port city (Chattogram), one in 2012 affecting 698 army personnel confirmed by third-generation Enzyme Immune Assay [17]. Another large one in 2018 resulted in 2800 cases [18]. A recent study conducted in a private laboratory in Dhaka identified small outbreaks of HEV infections throughout the year [19].

## 3. Classification, Reservoirs, and Transmission

Members of the species *Paslahepevirus Balayani* (previously known as *Orthohepevirus A)* have been assigned to eight different genotypes (HEV 1-HEV8) [1,20] with a diverse host range, being found in humans, as well as in a wide range of domestic and wild mammals (pig, wild boar, cow, deer, rabbit, camel) [1]. The host characteristics of HEV-1 to HEV-8 with a specific geographical distribution are shown in Table 1. The emerging zoonosis of HEV in rats (currently belonging to species *Rocahepevirus Ratti* changed from previous species *Orthohepevirus C*) causing acute hepatitis reported from Spain even among immunocompetent individuals [21] and transmission of Rat HEV Infection to humans in Hong Kong [22] reflect the importance of surveillance of these animals, as the chance of infecting humans in more expansive geographic areas globally, may occur in the future.

There are six major routes of HEV transmission:

(1) Waterborne transmission is most common for HEV-1 and HEV-2. It occurs via the fecal–oral route due to fecal contamination of drinking water through sewage, floods, heavy monsoon rain, overcrowding, and polluted water sources in refugee camps and slums [10].

In Bangladesh, all the outbreaks were due to fecally contaminated water supply in the rainy season or hot summer months.

(2) Foodborne transmission is associated with HEV-3 and HEV-4, which usually occurs through the ingestion of undercooked or raw body parts from an infected animal (e.g., liver, intestine, and meat of swine, natural wild boar bile juice, game meats such as wild deer, boar, and the hare, etc.) [23].

Transmission via this route is uncommon in Bangladesh.

(3) Zoonotic transmission other than foodborne occurs via contact with infectious body fluids or stool and occurs for HEV-3 and HEV-4. Although HEV-4 is widespread in many animals in India, such as domestic pigs, sheep, goats, and buffalo [24], the role of zoonotic transmission seems irrelevant in the context of human disease in India and Bangladesh. as here, HEV infections are caused uniformly by HEV-1 [25]. Swine are also a typical HEV-3 reservoir in developed countries, and contact creates a risk for HEV transmission in humans, which has been seen in Norway, where zoonotic transmission of HEV-3 and HEV-4 has occurred [26]. Farming cows, goats, and ducks are common (pigs uncommon due to the Muslim faith) in rural Bangladesh, and physical contact with these animals occurs during cleaning, catering, or milking. In the rural communities of Bangladesh, animal feces (cow dung) are commonly used as cooking fuel and for repairing house walls constructed with mud. Thus, exposure remains due to close contact with animals, and their feces occurs in rural communities.

HEV-8 was already found among Bactrian camels in China, but its zoonotic potential was unclear [27]. A recent study in multiple provinces in China investigated the cross-species transmission of HEV-8 from Bactrian camels to cynomolgus macaques. It demonstrated that HEV-8 could cause both acute and chronic HEV infections in cynomolgus macaques, a surrogate for humans, highlighting the potential risk of HEV-8 zoonotic transmission [28].

There is minimal research on the zoonotic transmission of HEV in Bangladesh. Bactrian camels are not native and domesticated dogs or cats as pets are not common in Bangladesh, unlike rats which are common pests [29]. Pigs in Bangladesh demonstrated HEV infection, and a study in three slaughterhouses in Dhaka found that the history of jaundice is significantly higher among pig handlers. Further investigation is needed to identify the HEV genotypes in pigs and pig handlers to understand the pig’s role in zoonotic HEV transmission in Bangladesh [30].

(4) Parenteral transmission of HEV through infected blood product transfusion has been reported in India (HEV-1) [31], Japan (HEV-3) [32], Germany [33], and Denmark [34].

A Bangladeshi study documented possible blood-borne transmission of sporadic HEV-1 in Dhaka from exposure to blood or blood-contaminated sharp instruments such as using recycled needle syringes and shaving in barbershops with unclean razors/blades [35].

(5) Vertical transmission: Infected mothers can transmit HEV to their fetus and infants via intrauterine [36] or transplacental route resulting in a higher risk of maternal and neonatal death [37]. During the 2008–2009 hepatitis E outbreak in Bangladesh, pregnant women with jaundice were at higher risk for miscarrying or delivering a stillborn baby than non-jaundiced pregnant women [38]. In addition, extrahepatic replication of HEV is possible in the human placenta, which may cause fetal and maternal mortality following acute liver failure [39]. The presence of HEV RNA in breast milk indicated a chance of transmission of HEV to a breastfed child from an infected mother but required additional research to fully understand this mode of transmission [40].

HEV-1 and HEV-2 have always been reported in pregnancy-related pathogenesis; however, a recent study on rhesus macaques infected with HEV-4 found the presence of HEV RNA in the liver, spleen, kidneys, and intestines of the deceased fetus. This evidence suggests the possible vertical transmission of HEV-4 partially due to the impairment and shifts in immune states during pregnancy [41].

(6) Person-to-person transmission: Epidemic disease due to HEV transmission via person-to-person contact was reported in Uganda from 2007 to 2008 and was strongly supported by Teshale et al. due to a high attack rate in households where multiple or single HEV-infected persons were residing, and no definite common source of infection (e.g., detection of HEV RNA in contaminated water) was found [42]. Considering the transmission mode is infrequent, the finding was opposed due to the lack of a detailed epidemiological assessment of the route of transmission, and the result remains controversial [43]. The person-to-person transmission of sporadic HEV-1 and HEV-2 was investigated by Khuroo et al. [44], who suggested the spread of HEV via household contact from the infected patient. However, the intrafamilial spread of sporadic HEV is very uncommon, and further evidence is required.

Other routes: Nosocomial transmission of HEV has been reported among hemodialysis patients and kidney transplant recipients [45]. Hepatitis E has also been detected in homosexual men indicating transmission of HEV can occur via sexual practices that involve the anus-hand-mouth [46].

## 4. Rohingya Refugee Camp in Bangladesh and HEV Scenarios

For over 900,000, Rohingya marked their 5th year of living in the southeastern part of Bangladesh, named “Cox’s Bazar” (Figure 1) in 2022 since their mass displacement from Myanmar [47]. This is undoubtedly one of the most crowdedly packed refugee camps in the developing world. Among them, 52% are women and girls, for whom HEV is a significant concern due to its higher fatality rate among pregnant women. Half of the HEV outbreaks in sub-Saharan Africa have occurred among refugees and displaced persons living in humanitarian crisis settings, including the recent outbreaks among refugees from Ethiopia in Sudan [48]. Open defecation and flooding, which occur in the camps, are additional risk factors for HEV emergence and can lead to contamination of nearby open sources of drinking water and food [49]. Water safety and sanitation are compromised in Cox’s Bazar refugee camp, where limited clean drinking water points cannot meet the increasing demands. Most water sources are not within 500 m of living places in the camp [50]. People dig holes for water and sometimes collect water for drinking, cooking, and other household chores directly from there, where people have bathed, washed, and practiced open defecation (Figure 2). The safe water scarcity in refugee camps deteriorates more during summer when temperatures in Cox’s Bazar can reach 40^0^ Celsius. During the rainy season, the camps are prone to contamination from landslides and flooding (Figure 3). Overall, the Cox’s Bazar refugee camp has enormous possibilities for disease spread via person-to-person, waterborne, and zoonotic transmission. This is exacerbated by the highly crowded living conditions where people are primarily non-immunized; have no separate living places for cattle; have an increased risk of sexual exploitation of men, women, and children; and a lack of sanitation and clean drinking water for both the community and household levels. From January to June 2018, 2253 cases of acute jaundice were recorded in the Cox’s Bazar refugee camp, raising the possibility of an HEV outbreak that could quickly spread in this susceptible population. However, a recent study investigating 275 blood samples collected from that outbreak through an enhanced surveillance system during less than a month (28 February to 26 March 2018) found that 56% of samples were positive for hepatitis A virus (HAV) compared to 0.4% for HEV diagnosed by anti-HEV ELISA for IgM with evidence of HAV-HEV co-infection [51]. It is important to note that sampling was conducted at selected sites for a brief period during this surveillance. Moreover, very few facilities were using rapid tests to confirm the clinical diagnosis of HEV. Thus, there is a possibility that many suspected cases were not sampled to test for HEV. Although the investigation did not point towards extensive HEV circulation, the presence of HEV and the similarity of transmission routes with HAV highlighted particular attention to continued vigilance of any flare-up of acute jaundice syndrome in the camps in the future.

## 5. Epidemiology

HEV epidemiology is divided into four unique zones (Figure 4) [10]. The reasons for these geographical differences in HEV case distributions have yet to be explored.

### 5.1. Hyperendemic and Endemic Zone

HEV-1 and HEV-2 hyperendemic zones are primarily in central, south and southeast Asia, Africa (east, west, and northern part), and Mexico. HEV is endemic in the Middle East, some regions of Southeast Asia such as Singapore, and several countries from South America (Ecuador, Brazil, Uruguay, and Argentina), as shown in Figure 4 [10].

During the last decade, HEV infection has increased among Latin American countries (620 million inhabitants living in 20 countries) with evidence of complex scenario of HEV epidemiology due to heterogeneity in the populations and geographical regions where HEV seroprevalence rates were reported between 4% and 40% [52]. Acute HEV cases reported in Latin America are mainly sporadic cases caused by HEV-3, but HEV-1 has also been detected. Only Mexico and Cuba have documented HEV-1, HEV-2 and HEV-3 outbreaks [53]. In Mexico, HEV-2 prevalence varied between 10–36% in adults with limited data on children~3% [54]. Nonetheless, there is a lack of information on HEV in animals and the environment in Latin America as it has only been studied in a few areas such as Argentina, Brazil, Uruguay, Costa Rica, Colombia, Venezuela, Bolivia, Mexico, and Cuba. Thus, there is a massive underestimation of the actual burden of HEV disease in Latin America [52].

Acute HEV cases are seen periodically throughout the year in Bangladesh, and the incidence rises in rainy seasons after floods, as it causes sewage contamination of piped and groundwater. A study from Bangladesh showed that the HEV incidence was 6% among >1200 people of all age groups [55]. Among pregnant women living in rural areas, the HEV infection rate is 46 cases/per 1000 person-year [56]. A prevalence survey among patients admitted to hospitals with hepatitis showed HEV caused 35% of sporadic acute hepatitis [57] and 22% of acute-on-chronic liver disease in Dhaka in 2008 [58]. An earlier survey from 2004 to 2006 identified HEV as the cause for 58% of general patients, 45% of pregnant patients with acute hepatitis, and 56% of cases with hepatic failure in Dhaka [19]. A serosurvey among adults in a central urban community in Dhaka showed that HEV prevalence is 30% among adults. An increasing trend of HEV prevalence has been observed correlating with an increase in age in urban Bangladesh with no gender difference [57]. The prevalence of HEV is higher in the urban versus rural population of Bangladesh [9]. The study that revealed 29 small HEV outbreaks (≥ 2 laboratory-confirmed cases) identified from a single private laboratory suggests that the actual burden of HEV infection in Bangladesh is much more significant than previously described by WHO using data from large outbreaks alone [19]. Strains of HEV-1a are dominant in Bangladesh and are associated with endemic outbreaks of HEV infection [59], while the 2018 Chattogram outbreak was caused by HEV-1f [60]. When compared with HEV-1a strains from India, Nepal, Pakistan, and Myanmar; within genotype 1a cluster, Bangladesh HEV strains formed a separate cluster with the 2010 HEV outbreak strains from northern Bangladesh; about 100% of the strains had A317T, T735I, L1120I, L1110F, P259S, V1479I, G1634K mutations associates AVH, FHF and RTF [59].

### 5.2. Distinctive Pattern Zone

The Hepatitis E distinctive zone is still limited to Egypt, as the infection pattern differs from other world regions [61]. In this zone, the prevalence is higher (60–80% seropositivity), especially in the first decade of life [62], and most young people develop HEV IgG antibodies with a seroprevalence resembling that of HAV. Interestingly, the incidence is lower among pregnant women, and the infection is either asymptomatic or very mild with no fulminant course. The HEV genotype infecting this population is HEV-1, with a peculiar subtype (subtype 1) not seen in Asia [63,64].

### 5.3. Sporadic Zone

Unlike the developing world, HEV is considered sporadic and rarely pathogenic in the developed world. However, HEV-3 and HEV-4 are increasingly recognized in developed countries [65]. Each year about 50–100 new cases of HEV are reported from Germany, France, the United Kingdom, and other countries from Europe among adults and children. Estimated annual HEV infections were 68,000 in France, 100,000 in the United Kingdom, and 300,000 in Germany [3]. HEV seroprevalence differs within the same country, between regions in developed countries, e.g., four times higher in south France than north France, but the reason has yet to explore [61]. More than two million new HEV infections are reported in the US yearly, and many infections are missing due to the unavailability of any FDA-approved diagnostic tests [61]. Sporadic cases of HEV infection through transfusion have been widely reported. Among blood donors, IgG seroprevalence was found: 33% in China; 29% in Germany; 27% in the Netherlands; 20% in Spain; 13 % in Austria; 12% in England; 6% in the USA, and 5% in both Ireland and Scotland [3]. The screening test for HEV in blood donors has been introduced in some countries but is still lacking in most regions [3]. It is also important to mention that in the developed world, HEV is sometimes misdiagnosed as another common liver disease in adults: drug-induced liver injury [66].

## 6. Clinical Presentation

The clinical presentation for acute HEV infection is like other acute viral hepatitis, generally asymptomatic (> 90% cases) or mildly symptomatic. During the first week of HEV infection, common symptoms are nausea and vomiting, malaise, fever, and body ache followed by dark-colored urine and jaundice among 5–30% of symptomatic patients; infection is self-limiting usually last <1 month [61,67]. From acute infection with no prior liver disease, progression to acute liver failure (ALF) is rare (~10%). Children with hepatomegaly and male patients over 60 years infected with HEV-3 and HEV-4 are susceptible to ALF, and mortality rate is high ~70% manifested with complications such as ascites, encephalopathy, and coagulopathy. Although after acute infection patients develop immunity but re-infection is possible [67].

In Bangladesh, among hospitalized HEV cases, the most common clinical features are jaundice, anorexia, nausea, vomiting, fever, and abdominal pain. Additionally, patients with an HAV or HBV co-infection are more prone to die, with an overall case fatality rate of 5% [9].

### 6.1. Pregnant Women

Pregnant women infected during third trimester with *HEV*-1 and HEV-2 are at a higher risk of having fulminant liver failure and a higher mortality rate of up to 20–25% [4]. In Bangladesh, there may be approximately 1000 maternal deaths per year associated with HEV [68]. Moreover, there is a higher risk of preterm deliveries with lesser survival rates for neonates following HEV infection, especially if infected during the third trimester of pregnancy, where 15–50% of infants die within a week of birth [69]. There are also higher chances of intrauterine death due to HEV. Another study conducted in Bangladesh demonstrated that pregnancy might be associated with an inverse correlation between IgG antibody response and viral load (higher viral loads and a lower IgG response) with acute HEV-1 infection leading to poorer outcomes with an acute HEV infection compared to non-pregnant women [70]

### 6.2. Chronic HEV Infection

Chronic HEV infection is considered when it persists for more than three months [61]. HEV-3 and HEV-4 as well as rat HEV can lead to chronic HEV and cirrhosis in immunosuppressed and solid organ transplant patients (both children and adults) from developed countries within 2–3 years of infection [61,71]. Patients with lymphopenia and receiving immunosuppressive drugs such as Tacrolimus are at high risk of developing chronic HEV infection [71].

### 6.3. HEV Superinfection

HEV infection in a patient with chronic liver disease (CLD) previously infected with another virus (Hepatitis B), is considered HEV superinfection. A study has shown that HEV superinfection accelerated disease progression and increased long-term mortality in patients with hepatitis B liver cirrhosis, especially in patients with end-stage liver cirrhosis [72].

## 7. Diagnosis

The incubation period of HEV varies from 15 to 60 days, and HEV RNA can be detected in feces and serum samples and persists for ~4 weeks in blood and ~6 weeks in stool after infection. Either acute or chronic HEV RNA detection and quantification in blood, stool, or other body fluids remain the golden standard for active HEV infection. Detection of anti-HEV antibodies, including anti-HEV IgG and IgM, also plays a vital role in diagnosing HEV. Still, immunosuppressed patients often showed negative results for serology due to low titer of anti-HEV antibodies. Positive results for anti-HEV IgM (with or without positive anti-HEV IgG) together with HEV RNA represent current acute infection; only a positive result for anti-HEV IgG represents past infection of HEV. Patients reinfected with HEV usually show a positive result for anti-HEV IgG and HEV RNA but not anti-HEV IgM. Diagnostic algorithm for HEV infection is shown in (Figure 5) [3,67].

## 8. Treatment

There are very few articles regarding the treatment of hepatitis E. To date, there are no approved drugs available for HEV treatment. Ribavirin (a Guanosine analog) is a possible treatment option for chronic HEV. It has been found to inhibit HEV replication in immunocompromised patients, thereby allowing HEV clearance, which can prevent fatal complications; still, there is also a history of treatment failure and additional studies are required to understand its mechanism of action [71]. It is important to note that ribavirin is teratogenic and, thus, contraindicated for pregnant women at the highest risk of HEV. In combination, as treatment options for chronic hepatitis E, ribavirin and sofosbuvir (a highly potent inhibitor of specific polymerase in the hepatitis C virus) have been tested after failing ribavirin monotherapy. Nonetheless, the combination therapy was proven inefficient in sustaining viral response among organ transplant patients with chronic HEV infection and showed several complications [73]. It is worth mentioning that these antivirals are too expensive, need to be taken for more extended periods, and are not affordable for the general population. Hence, alternative innovative treatment options are urgently required for pregnant women and chronic HEV infection patients failing ribavirin or other antiviral treatment.

## 9. Vaccination

HEV 239 (Hecolin^®^), manufactured by Xiamen Innovax, China, is the only commercialized vaccine available in China since 2012 as a three-dose regimen (0, 1 month, and six months) and has recently been available in Pakistan. Hecolin^®^ is composed of virus-like particles expressed in *Escherichia coli* (368–606 aa of ORF2) from a Chinese strain of HEV-1. The vaccine also provides cross-protection against different genotypes, preventing HEV-1 (the vaccine genotype) and HEV-4 (the most prevalent genotype in China) infections and can provide sustained protection for over four years (87% efficacy at 4.5 years) [74]. However, for WHO prequalification and a global launch of this vaccine, more safety data need to be generated, especially in other populations at risk, such as pregnant women, children <16 years, older patients, patients with chronic liver disease, solid organ transplant recipients, HIV and those with immune disorders. Overall, the vaccine has already been tested in Phase 1 and 2 pre-licensure trials and a large Phase 3 trial among healthy participants, where the safety data was reassuring. Considering the importance of determining the vaccine’s safety, immunogenicity and efficacy outside China, where HEV-1 infection is highly endemic, our team conducted a Phase IV effectiveness trial of the HEV 239 vaccine in rural Bangladesh among 20,000 women aged 16–39 years which has been completed recently [75]. The data will help to inform policymakers on HEV vaccination strategy shortly.

## 10. Knowledge Gap and Recommendations

The panorama of HEV research is changing drastically with new and frequent developments in the field, yet several knowledge gaps still need to be addressed. The pathogenic mechanism of HEV is still unclear. Does HEV confer lifelong immunity after infection, or how long immunity persists; we do not know. There is no documentation on long-term complications among chronic HEV patients.

The safety of Hecolin^®^ when used concomitantly with other vaccines such as Hepatitis B or Human Papilloma Virus vaccines (HPV) needs to be investigated. The coadministration of the HEV vaccine with HPV to young adolescent girls before they enter their child-bearing years would be the most attractive vaccination option, as maternal immunization could be impractical given the 7-month spacing between the entire course of three doses of vaccine. This would also cater to any suspected waning immunity over the long term. Hecolin^®^ vaccine has never been used in an outbreak setting; there is a shortage of much-needed data about how the vaccine might perform in an outbreak. Spacing three doses of vaccine over six months pose a challenge to the access and use of the vaccine in routine immunization; hence, whether a single dose of the vaccine could provide sufficient protection to prevent infection or to control expansive or prolonged outbreaks needs to be explored. Most importantly, vaccination should be investigated among pregnant women. No safety data have been generated yet in this most vulnerable population. Generally, most vaccines, except live ones, can be safely administered to pregnant women. Although Hecolin^®^ contains thiomersal, there is no evidence that thiomersal-containing vaccines cause adverse effects in offspring of women who received these vaccines during pregnancy. Immunoglobin G elicited by the vaccination to pregnant women is transferred to the fetus via placenta. Additionally, mucosal immunoglobulins are secreted into the colostrum and milk, which protect the newborn in the postpartum period. Additionally, thus, acquired immunity by vaccination benefits not only mothers but for fetuses and neonates.

To reduce the global burden, strategies should focus on health education, improving water, sanitation, and hygiene in endemic regions, rapid and accurate diagnostics, appropriate treatment, vaccine development, and prevention. All patients presenting with a compatible clinical and laboratory picture of acute hepatitis with no evidence of an apparent cause of liver damage should be tested for HEV. To achieve this, we must train healthcare workers to mitigate the lack of awareness of HEV and increase access to testing services by establishing Point-of-Care (POC) or with-patient testing by the rapid test method, which will allow physicians and medical staff to accurately achieve real-time, laboratory-quality diagnostic results within minutes rather than hours, especially during outbreaks. There are some commercial diagnostic kits with high sensitivity and specificity, but they are not available or affordable in the developing world. Identifying environmental factors like safe water, sanitation and hygiene, farming, and good food habits, may help to reduce transmission. Research on HEV involving pregnant and non-pregnant women, children, and immunosuppressed individuals may help us better appreciate the global burden of this dreaded disease. Clear guidance is needed on how blood and organ donors will be screened for HEV to avoid HEV transmission through transfusion or organ transplantation.

## 11. Conclusions

Our understanding of HEV has changed entirely over the decades. However, many enigmas around HEV need to be solved, which require special attention due to its unique epidemiologic patterns, limited treatment options, impact on low-resource and displaced communities, and disproportionate disease burden in pregnant women. Undoubtedly, HEV must be eliminated to achieve WHO’s recommendation to eradicate infectious hepatitis by 2030. However, without the use of a vaccine to protect high-risk populations from morbidity and mortality, it will be impossible.

## Figures and Tables

**Figure 1 viruses-15-00063-f001:**
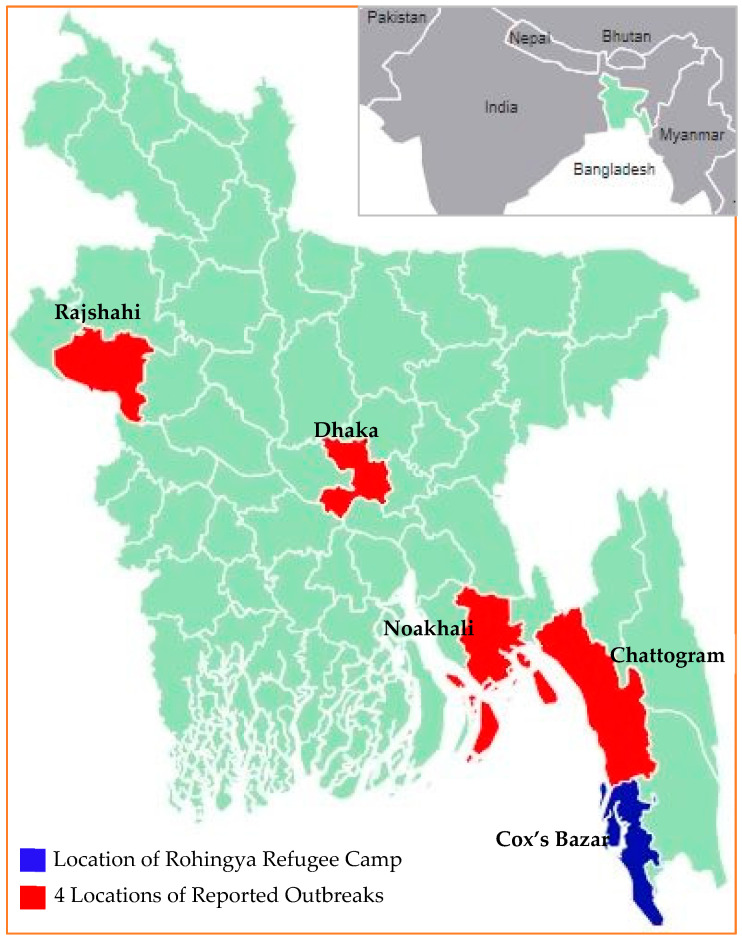
Map showing reported outbreak and refugee camp locations in Bangladesh.

**Figure 2 viruses-15-00063-f002:**
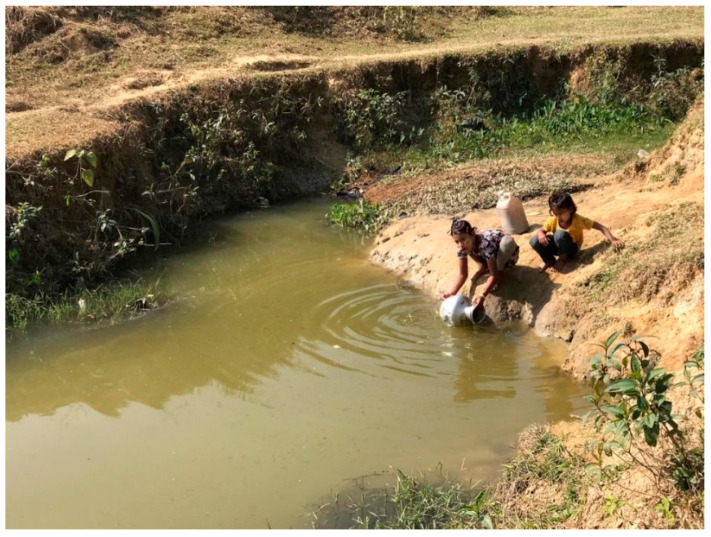
Water source in Rohingya refugee camp (photo was taken with Permission).

**Figure 3 viruses-15-00063-f003:**
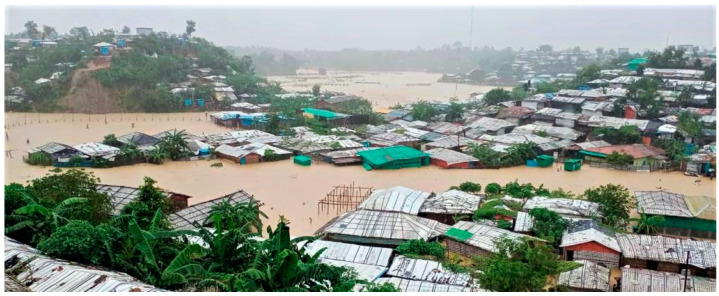
Rohingya refugee camp Bangladesh.

**Figure 4 viruses-15-00063-f004:**
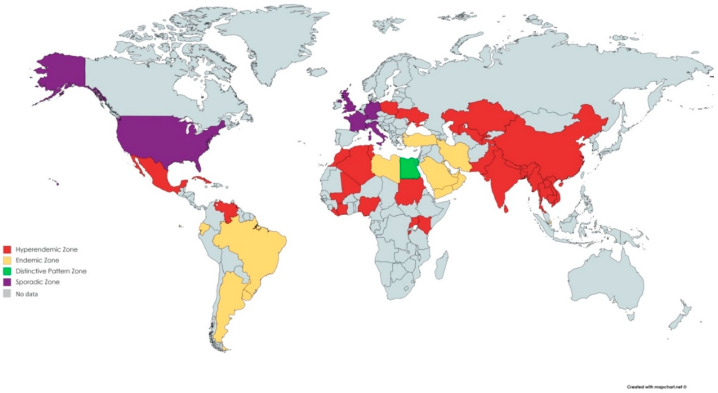
Hepatitis E epidemiology in 4 distinctive zones.

**Figure 5 viruses-15-00063-f005:**
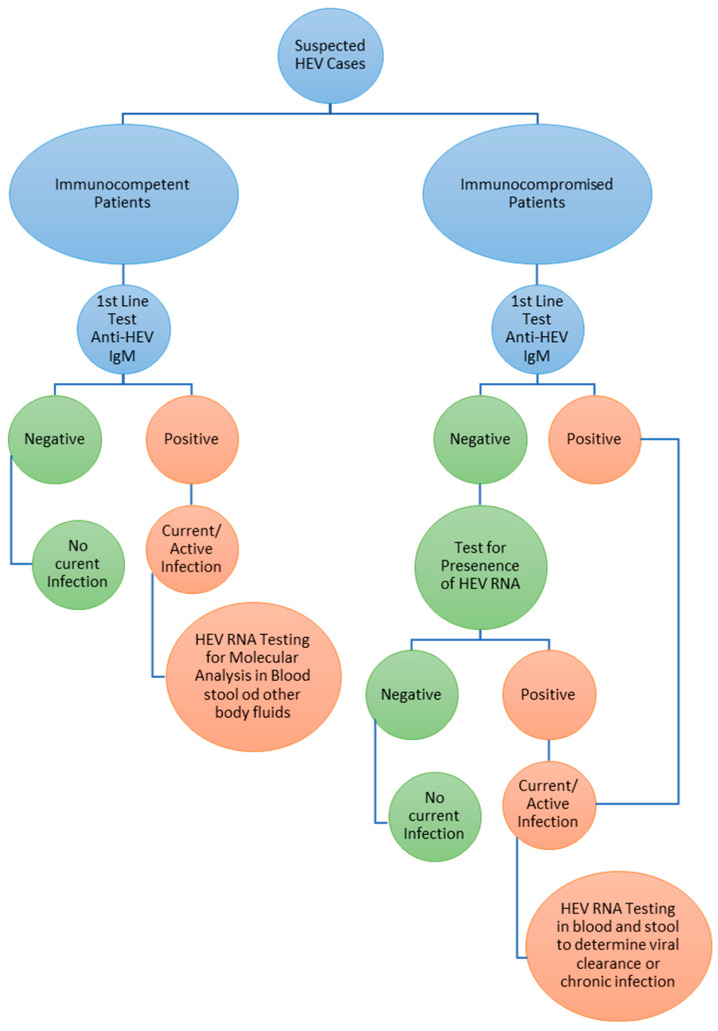
Diagnostic algorithm of HEV infection [3].

**Table 1 viruses-15-00063-t001:** Geographical distribution of HEV genotypes and their hosts.

Genotypes	Host	Distribution
HEV-1	Human Only (Experimental: higher primates such as Rhesus monkeys)	Asian LMIC (Myanmar, Pakistan, Bangladesh, India, and Nepal), Mexico, the Middle East and Africa
HEV-2		
HEV-3	Humans, domestic pigs, wild boars, deer and other mammalian animals: rodents, rabbits, cattle, horses, goats, sheep, dogs, cats, mongoose, bottlenose dolphin	Predominant in industrialized but also in source-limited countries Asia, Europe, the Netherlands, France, Germany, Taiwan, Greece, France, Spain, Germany, the UK, Kyrgyzstan, Italy, Uruguay, New Zealand, South America, Canada, Australia, Mexico Switzerland
HEV-4	Humans, and mammals: domestic pigs, wild boars, cattle, goats, sheep, deer, leopard, black bear, yak	Affluent Asian countries including China, Taiwan, Vietnam, and Japan; also circulating in Europe
HEV-5 and HEV-6	Wild boars only	Japan
HEV-7	Dromedary Camels, Human	Middle East, Africa, Saudi Arabia
HEV-8	Bactrian Camels	China

## Data Availability

Not applicable.

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
