# Peer review of "Hepatitis E Virus (HEV) Synopsis: General Aspects and Focus on Bangladesh"

_viruses, 2022, doi:10.3390/v15010063_

Round 1

Reviewer 1 Report

General comments:

The title should mention that the synopsis focuses in Bangladesh. A possible title could be: “Hepatitis E Virus (HEV) Synopsis: general aspects and focus on Bangladesh”.

Including a map of Bangladesh (as a figure), showing the cities and places mentioned in the manuscript (such as the refugee camp) should clarify the regions mentioned in the paper to readers unfamiliar with the country.

Specific comments:

Line 21: Although the hepatitis E virus is usually named as such, its official name (given by ICTV) is Paslahepevirus balayani. This information should be included in the manuscript, together with the family and genus/subgenus to which it belongs. This can be added to the first line of the introduction:

Moreover, the recent new nomenclature owes its name to the virologist who discovered the virus (Balayan), who is mentioned latter in the text (line 68).

Line 86: genotype is abbreviated as GT throughout the manuscript, but frequently HEV genotypes are named: HEV-1 to HEV-8 in the literature. Please change this in all the paper (so that it is clearer and more homogeneous with the rest of the published works).

Line 91: the routes of HEV transmission are named.

Some routes of transmission are mainly related to some particular genotypes, and this should be mentioned in this part of the manuscript. For example: in points 1 and 2, HEV-1 and HEV-2 are mostly transmitted through contaminated water, while zoonotic transmission occurs for HEV-3 and HEV-4.

Line 150: It is stated: “Hepatitis E has also been detected in homosexual men indicating transmission of HEV can occur via a sexual route [38]”. However, the reference number 38 does not contain this information; it is possible that the correct reference is number 37. Please, provide the correct citation for this and check the rest of the references. On the other hand, regarding the statement, the reference cited does not mention “sexual transmission”. It is possible that HEV transmission reported on the cited paper is not through the classic "sexual" route (by contact of mucous membranes and fluids), but that transmission of the virus can occur through sexual practices that involve anus-hand-mouth (this is clearly stated in the reference). Therefore, affirming that HEV presents "sexual transmission" would not be entirely correct. Please, rephrase this statement.

Lines 181 to 185: Please, provide a reference or references for the worldwide prevalence information.

Line 184: Table 2 repeats the information of Figure 3. Please, delete it. 

Lines 183 to 203: Only the endemic scenario in Bangladesh is explained. Because it is a review article, it would be necessary to make a brief explanation of other endemic areas of the world, such as Latin America.

Line 206: please, clarify what do you mean by “high prevalence” (in Egypt). Giving percentages would help to understand this statement.

Line 216: Please, check parenthesis.

Lines 213 to 218: Only some prevalence percentages (for some countries) are expressed, but many study sites are missing (Spain for example). Please expand the information in this section.

Line 222: The sentence “The clinical presentation of HEV varies according to geographical location…” is not correct, please rephrase it.

Line 234: Chronic hepatitis E has also been described due to HEV-4 (not only HEV-3) (see WHO 2022: https://www.who.int/es/news-room/fact-sheets/detail/hepatitis-e).

Line 253: in line 253 the authors stated: “In developed countries, chronic HEV infections (GT 3, 4, and 7) are…” while in line 234 they said that chronic hepatitis E is due only to HEV-3. Please, change these sentences with the correct information.

Lines 253 to 259: the sentence “In developed countries, chronic HEV infections (GT 3, 4, and 7) are more common in the asymptomatic form in otherwise healthy persons” is not correct. HEV infection by genotypes 3 and 4 in immunosuppressed individuals can progress to chronicity, this is mostly reported in developed countries. But chronic infections are not asymptomatic, this is not reported as far as I know.

Lines 260 to 268: HEV infection in a patient with chronic liver disease (CLD) is not always a superinfection. The only case in which it could be said that it is a superinfection is in those patients previously infected with another virus, as in the case of hepatitis B that is mentioned. But this is not entirely clear in the paragraph. Please clarify. One can mention on one hand, HEV superinfection (in individuals previously infected with another agent), and, on the other hand, HEV infection in individuals with CLD (for different causes, not always infectious).

Line 272: Please change “see” by “detect”.

Line 357: please, rewrite the number 20,0000 correctly.

Lines 360 to 375: the information included in these lines are not Conclusion. These are topics that deserve more information, bibliographic support and discussion, and could be included in the body of the manuscript.

Table 1: please change GT… by HEV-1 to HEV-8.

Author Response

On behalf of my co-authors and myself, I am submitting the revised manuscript entitled “Hepatitis E Virus (HEV) Synopsis: general aspects and focus on Bangladesh” for publication in Viruses-MDPI. Thanks for the constructive comments on our manuscript, which helped improve the quality of our submitted manuscript.

Please see our responses below:

Reviewer 1:

  • The title should mention that the synopsis focuses in Bangladesh. A possible title could be: “Hepatitis E Virus (HEV) Synopsis: general aspects and focus on Bangladesh”.

Response: Thanks for your valuable suggestion. We have revised the title as suggested (Line 2-3).

  • Including a map of Bangladesh (as a figure), showing the cities and places mentioned in the manuscript (such as the refugee camp) should clarify the regions mentioned in the paper to readers unfamiliar with the country.

Response: Thanks. We agreed and added a map as Figure 1 showing four locations of the reported outbreaks in Bangladesh and Rohingya Refugee camps in Cox’s Bazar.

Specific comments:

  • Line 21: Although the hepatitis E virus is usually named as such, its official name (given by ICTV) is Paslahepevirus balayani. This information should be included in the manuscript, together with the family and genus/subgenus to which it belongs. This can be added to the first line of the introduction:

Moreover, the recent new nomenclature owes its name to the virologist who discovered the virus (Balayan), who is mentioned latter in the text (line 68).

Response: Thanks. This information is added in the first line of the introduction (Line 36-39).

  • Line 86: genotype is abbreviated as GT throughout the manuscript, but frequently HEV genotypes are named: HEV-1 to HEV-8 in the literature. Please change this in all the paper (so that it is clearer and more homogeneous with the rest of the published works).

Response: The whole paper is revised accordingly for a clearer and more homogeneous description as advised.

  • Line 91: the routes of HEV transmission are named.

Some routes of transmission are mainly related to some particular genotypes, and this should be mentioned in this part of the manuscript. For example: in points 1 and 2, HEV-1 and HEV-2 are mostly transmitted through contaminated water, while zoonotic transmission occurs for HEV-3 and HEV-4.

Response: Thanks. We have now mentioned this in lines 119 and 126.

  • Line 150: It is stated: “Hepatitis E has also been detected in homosexual men indicating transmission of HEV can occur via a sexual route [38]”. However, the reference number 38 does not contain this information; it is possible that the correct reference is number 37. Please, provide the correct citation for this and check the rest of the references. On the other hand, regarding the statement, the reference cited does not mention “sexual transmission”. It is possible that HEV transmission reported on the cited paper is not through the classic "sexual" route (by contact of mucous membranes and fluids), but that transmission of the virus can occur through sexual practices that involve anus-hand-mouth (this is clearly stated in the reference). Therefore, affirming that HEV presents "sexual transmission" would not be entirely correct. Please, rephrase this statement.

Response: Thanks for pointing out this. We have corrected the reference (new reference 46) and rephrased the statement (Line 193-197).

  • Lines 181 to 185: Please, provide a reference or references for the worldwide prevalence information.

             Response: We added reference 10.

  • Line 184: Table 2 repeats the information of Figure 3. Please, delete it.

Response: Table 2 is deleted in the revised version.

  • Lines 183 to 203: Only the endemic scenario in Bangladesh is explained. Because it is a review article, it would be necessary to make a brief explanation of other endemic areas of the world, such as Latin America.

              Response: We added information on Latin American Region as suggested (Line 252-263)

  • Line 206: please, clarify what do you mean by “high prevalence” (in Egypt). Giving percentages would help to understand this statement.

Response: HEV seropositivity among Egyptians was 60–80%, especially in the first decade of life. Added this as suggested with new reference [62. Sayed IM, Abdelwahab SF. Is Hepatitis E Virus a Neglected or Emerging Pathogen in Egypt? Pathogens. 2022; 11(11):1337. https://doi.org/10.3390/pathogens11111337)] (Line 292-293).

  • Line 216: Please, check parenthesis.

             Response: Corrected the typos all over the articles.

  • Lines 213 to 218: Only some prevalence percentages (for some countries) are expressed, but many study sites are missing (Spain for example). Please expand the information in this section.

Response: This section has been revised completely as suggested. (Line 300-317)

  • Line 222: The sentence “The clinical presentation of HEV varies according to geographical location…” is not correct, please rephrase it.

Response: We deleted the sentence.

  • Line 234: Chronic hepatitis E has also been described due to HEV-4 (not only HEV-3) (see WHO 2022: https://www.who.int/es/news-room/fact-sheets/detail/hepatitis-e).

Response: We deleted the sentence as it’s not a correct statement; added a separate section on chronic HEV infection with this reference included (Line 372-383)

  • Line 253: in line 253 the authors stated: “In developed countries, chronic HEV infections (GT 3, 4, and 7) are…” while in line 234 they said that chronic hepatitis E is due only to HEV-3. Please, change these sentences with the correct information.

Response: As mentioned above we revised the chronic HEV infection section with the correct statement (Line 372-383)

  • Lines 253 to 259: the sentence “In developed countries, chronic HEV infections (GT 3, 4, and 7) are more common in the asymptomatic form in otherwise healthy persons” is not correct. HEV infection by genotypes 3 and 4 in immunosuppressed individuals can progress to chronicity, this is mostly reported in developed countries. But chronic infections are not asymptomatic, this is not reported as far as I know.

Response: As mentioned above we revised the chronic HEV infection section with the correct statement (Line 372-383)

  • Lines 260 to 268: HEV infection in a patient with chronic liver disease (CLD) is not always a superinfection. The only case in which it could be said that it is a superinfection is in those patients previously infected with another virus, as in the case of hepatitis B that is mentioned. But this is not entirely clear in the paragraph. Please clarify. One can mention on one hand, HEV superinfection (in individuals previously infected with another agent), and, on the other hand, HEV infection in individuals with CLD (for different causes, not always infectious).

Response: We revised the HEV superinfection section with the correct statement (Line 385-389)

  • Line 272: Please change “see” by “detect”.

             Response: We revised the whole diagnostic section (Line 398-409)

  • Line 357: please, rewrite the number 20,0000 correctly.

Response: The Typo has been corrected now.

  • Lines 360 to 375: the information included in these lines are not Conclusion. These are topics that deserve more information, bibliographic support and discussion, and could be included in the body of the manuscript.

Response: We changed these sections to Knowledge Gap and Recommendations and added a conclusion section in the revised manuscript (Line 503-573)   

  • Table 1: please change GT… by HEV-1 to HEV-8.

Response: As mentioned above, we revised accordingly for a clearer and more homogeneous description

Thank you!

Reviewer 2 Report

Hepatitis E virus (HEV) has become a significant public health issue in recent years. In this review, the authors have summarized the recent developments in epidemiology, clinical characteristics, diagnosis, and treatment of HEV, with a special focus on Bangladesh. Overall the review article has captured a good amount of information. However, this article should have covered several critical recent developments in HEV. 

Here is the list of points; please include them in the appropriate places. 

  1. The authors have mentioned genotype-1 to genotype-8 of HEV, which belongs to Orthohepevirus A. However, other species of HEV have also been reported to infect humans. Examples: Rat Hepatitis E Virus Linked to Severe Acute Hepatitis in an Immunocompetent Patient (PMID: 30649379), Hepatitis E Virus Species C Infection in Humans, Hong Kong (PMID: 34718428), Orthohepevirus C infection as an emerging cause of acute hepatitis in Spain: First report in Europe (PMID: 35167911).
  2. Genotype-8 has been reported only in Bactrian camels. However, a study investigated the cross-species transmission of genotype-8 HEV in cynomolgus macaques (PMID: 30700602).
  3. Pregnancy-associated pathogenesis was always reported for genotype-1 and genotype-2. However, a recent study has reported a vertical transmission of genotype-4 HEV in rhesus macaques (PMID: 33060782). 
  4. A recent study reported the prevalence of HEV antibodies from a nationally representative serosurvey in Bangladesh. (PMID: 34549775)
  5. Another study reported IgG immune responses and hepatitis E virus (HEV) viral load and their associations with pregnancy in Bangladesh. PMID (33460834)
  6. Lines 175-178 Regarding antibody testing in serums collected in 2018 and 2017. However, the cited article (40) was published in 2009. In this context, it was reported in PMID: 33914782 that HAV, not HEV was the primary concern in Cox's bazar among Rohingya refugees. Could authors explain this?
  7. A very recent study reported the epidemiological profile of a hepatitis E virus outbreak in 2018, Chattogram, Bangladesh. PMID (36006262)
  8. The authors can also discuss HEV subgenotypes. Examples: A tightly clustered hepatitis E virus genotype 1a is associated with endemic and outbreak infections in Bangladesh PMID (34293039) and Hepatitis E virus genotype 1f outbreak in Bangladesh, 2018 PMID (33331650). Within the genotype 1a cluster, Bangladesh HEV strains formed a separate cluster from the 2010 HEV outbreak strains from northern Bangladesh. 80.9 to 100% of the strains had A317T, T735I, L1120I, L1110F, P259S, V1479I, G1634K mutations associates AVH, FHF, and RTF PMID (34293039).  
  9. There has been a significant improvement in the HEV field in recent years. Hence, the authors should include recent publications in their review. 

Round 2

Reviewer 2 Report

The authors have addressed all my concerns and revised the manuscript carefully.